# Atomic-level polarization reversal in sliding ferroelectric semiconductors

Fengrui Sui [1], Haoyang Li[1], Ruijuan Qi [1,2] ✉, Min Jin[3] ✉, Zhiwei Lv[1], Menghao Wu[4], Xuechao Liu[5], Yufan Zheng[1], Beituo Liu[1], Rui Ge [1], Yu-Ning Wu [1] ✉, Rong Huang [1], Fangyu Yue [1,6,7] ✉, Junhao Chu[1,8] & Chungang Duan[1,6,7]

Intriguing "slidetronics" has been reported in van der Waals (vdW) layered non-centrosymmetric materials and newly-emerging artificially-tuned twisted moiré superlattices, but correlative experiments that spatially track the interlayer sliding dynamics at atomic-level remain elusive. Here, we address the decisive challenge to in-situ trace the atomic-level interlayer sliding and the induced polarization reversal in vdW-layered yttrium-doped γ-InSe, step by step and atom by atom. We directly observe the real-time interlayer sliding by a 1/3-unit cell along the armchair direction, corresponding to vertical polarization reversal. The sliding driven only by low energetic electron-beam illumination suggests rather low switching barriers. Additionally, we propose a new sliding mechanism that supports the observed reversal pathway, i.e., two bilayer units slide towards each other simultaneously. Our insights into the polarization reversal via the atomic-scale interlayer sliding provide a momentous initial progress for the ongoing and future research on sliding ferroelectrics towards non-volatile storages or ferroelectric field-effect transistors.

In the big data era, the demand for data processing and storage manifests a booming growth. Novel in-memory computing nanodevices based on ferroelectric field-effect transistors that successfully combine ferroelectricity and semiconductor's properties hold great potential to overcome the data traffic bottleneck relying on the von Neuman frame. Such devices require simultaneous optimizations of high ferroelectric polarizations with low switching energy barriers, moderate bandgaps with high mobility, etc[1]. In recent years, the intriguing sliding ferroelectricity with low barriers has been widely and solely discovered in two-dimensional (2D) materials[2–6], whose out-of-plane (OOP) polarization reversal originates from the in-plane interlayer sliding[1,3,7–10]. It expands the 2D ferroelectrics from the few polar candidates to most known 2D materials that have suitable bandgaps and high mobility, including the high-performance semiconductors like MoS$_2$[11], InSe[12], and GaSe[13]. Recently, sliding induced multiple polarization states manipulation in multi-layered 2D sliding ferroelectrics[2,11,14] makes them promising systems for neuro-inspired computing applications. Hence, understanding the sliding mechanism is of fundamental importance to promote the researches on the sliding-related physics and the applications of slidetronics with high-speed and low-consumption. So far, many breakthroughs have been achieved not only on the constructions of twisted moiré superlattices

[1]Key Laboratory of Polar Materials and Devices (MOE), School of Physics and Electronic Science, East China Normal University, Shanghai 200062, China. [2]National Key Laboratory of Materials for Integrated Circuits, Shanghai Institute of Microsystem and Information Technology, Chinese Academy of Sciences, Shanghai 200050, China. [3]College of Materials, Shanghai Dianji University, Shanghai 201306, China. [4]School of Physics, Huazhong University of Science and Technology, Wuhan 430074, China. [5]Shanghai Institute of Ceramics, Chinese Academy of Sciences, Shanghai 200050, China. [6]Collaborative Innovation Center of Extreme Optics, Shanxi University, Taiyuan, Shanxi 030006, China. [7]Shanghai Center of Brain-inspired Intelligent Materials and Devices, East China Normal University, Shanghai 200062, China. [8]National Laboratory of Infrared Physics, Shanghai Institute of Technical Physics, Shanghai 200083, China. ✉e-mail: rjqi@ee.ecnu.edu.cn; jmaish@aliyun.com; ynwu@phy.ecnu.edu.cn; fyyue@ee.ecnu.edu.cn

showing sliding ferroelectricity[8,9,15] but also on the polarization reversal dynamics based on theoretical predictions[11,16,17]. Especially, the twisted lattice induced antiferroelectric moiré domains alternative changing behavior (indirect phenomena related to the domain wall motion) has been observed at micrometer-scale through electron microscopy or atomic force microscopy (AFM) including (Kelvin) piezo-force microscopy (K/PFM) characterizations[10,18]. Nonetheless, the real-time manipulation and investigation on the interlayer sliding induced polarization reversal at atomic level[19] are still in lack.

Although the spherical aberration corrected transmission electron microscopy (Cs-TEM) is regarded as the most straightforward approach to probe and confirm the sliding mechanism by in-situ monitoring the atomic interlayer sliding dynamics, atomic-scale in-situ observation of the interlayer sliding in 2D vdW-layered ferroelectrics remains elusive because of the mismatch between the TEM technique and the ultrathin feature of 2D layered ferroelectric structures[20] (Supplementary Fig. 1, Supplementary Note 1). On one hand, the high-resolution TEM (HRTEM) or high-angle annular dark field scanning TEM (HAADF-STEM) requires the focused ion beam (FIB)-prepared cross-sectional lamella to have a relatively flat and even area with a specific zone axis (Supplementary Fig. 1h). On the other hand, most 2D vdW-layered sliding ferroelectrics are atomic-scale few-layered systems with relatively low stiffness, making it a prohibitive challenge to obtain a suitable lamella for atomic imaging analyses. In other words, direct observation of the interlayer directional sliding dynamics prerequires thicker or three-dimensional vdW-based materials or structures (Supplementary Fig. 1h), because the real-time in-plane interlayer sliding at atomic-scale under the OOP electrical field can only be directly observed from the cross-section profile of the 2D layered specimen. Therefore, it is intrinsically distinguished from the reported plane-view observation of the moiré patterns in bi- or few-layered twisted structures, and also different from the already succeeded cases in conventional ferroelectric materials with evident phase changes or domain formation/motion at microscale[21–26]. Recently, the robust ferroelectricity in γ-InSe semiconductor was activated by yttrium-doping (InSe:Y) from few-layer to bulk with an adjustable bandgap around 1.2 eV and a distinguished mobility of 2000 cm$^2$/V·s[12]. This vdW semiconductor with a long-thickness-ordered ferroelectricity provides a substantial platform for exploring the interlayer sliding behavior and resulted polarization inversion mechanism by the Cs-TEM technique.

Here, by utilizing the atomic-scale in-situ biasing Cs-TEM, we directly observe the reversible "ripplocation" domain walls[1,27] formation/motion and the external electric field induced layer stacking change in vdW-layered ferroelectric InSe:Y semiconductor. Our in-situ investigations combining with conductive atomic force microscopy (C-AFM) results demonstrate a stepwise ferroelectric response related to multiple polarization states transition. Furthermore, the kinetics of atomic-scale interlayer sliding in InSe:Y is clearly observed under the electron-beam (e-beam) illumination in STEM, which induces local electrostatic charging and local electric field[21,25] for polarization switching in ferroelectrics[26,28,29]. Such sliding is observed excluding the possible influence from the electrode, further confirming the OOP polarization switching dynamics. Additionally, first-principles calculations provide an interpretation of the role of doped-Y for observing sliding dynamics, and more importantly, validate the observed preferred reversal pathway of the simultaneous relative sliding of bilayers as blocks in InSe:Y with a sliding barrier of ~31.0 meV/f.u.

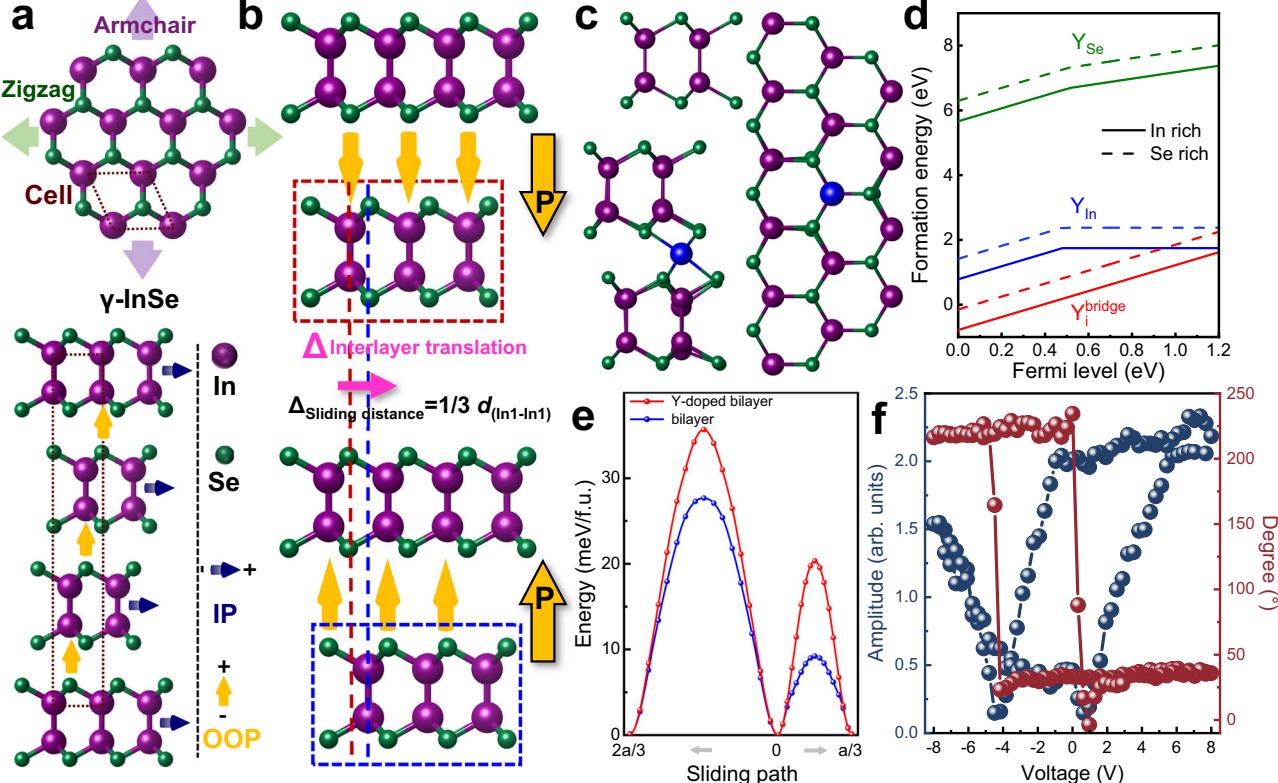

**Fig. 1 | The role of Y-doping in InSe for ferroelectricity. a** Structure schematics of InSe single layer (top-view, upper) and the atomic arrangement (side-view, lower) of γ-InSe, demonstrating the in-plane (dark blue arrows) polarization along the AC direction and the OOP (orange arrows) polarization along the thickness (or *c*)-direction. **b** Schematical atomic arrangements of the OOP polarization reversal in InSe by interlayer sliding. **c** Energetically favorable sites of doped-Y (blue dots) in InSe lattices, interstitial or substituting. **d** The formation energies of Y-related defects predicted from first-principles calculations under In or Se-rich conditions. **e** The influence of doped-Y on the interlayer sliding energy barriers for bilayer InSe (right for the short path). **f** Local PFM amplitude and phase loops during the polarization switching process in InSe:Y (~18 nm).

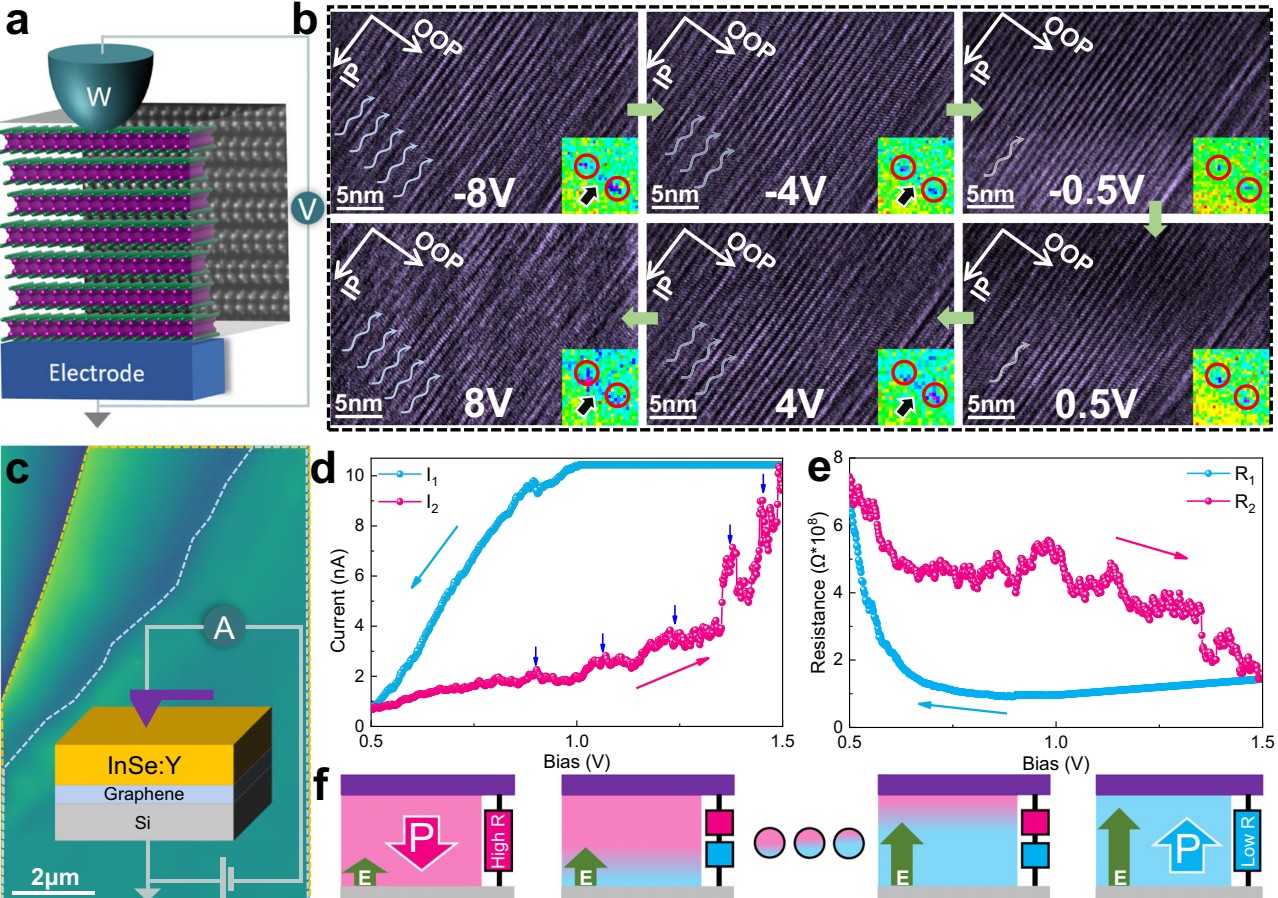

**Fig. 2 | In-situ biasing-induced interlayer sliding in InSe:Y by HRTEM and the I-V curves during the polarization switching by CAFM. a** Schematic of the in-situ biasing system providing the OOP electric field. **b** HRTEM snapshots during in-situ biasing TEM analyses at different biasing voltages with the FFT patterns in insets. **c** The AFM image of InSe:Y flake for CAFM measurements (see the inset). **d** The I-V loop from CAFM measurements. **e** The corresponding bias-dependent resistance (R-V) curve from (**d**). The pink arrows in (**d**) indicate current jumps as the voltage increases (the pink line), corresponding to the abrupt resistance shifts in (**e**). **f** The scheme for different (or multiple) polarization states during polarization switching.

## Results

### The role of Y-doping in the ferroelectricity in γ-InSe

Rhombohedral γ-InSe (lattice parameters $a = b = 4.00$ Å and $c = 25.32$ Å) belongs to the $C^5_{3v}$ ($R3m$) non-centrosymmetric space group with ABC-style stacking, possessing three armchair (AC) and three zigzag (ZZ) directions (Fig. 1a, upper). This specified symmetry-broken atom-arrangement, which can be characterized by the second-harmonic generation (Supplementary Fig. 2), illustrates the OOP polarization due to the vertical interlayer dipole moment between the bottom In atoms in one layer and the Se atoms in the adjacent layer (Fig. 1a, lower)[30]. It has also been theoretically predicted to possess sliding ferroelectricity[6,15] if a vertical electric field can induce the interlayer sliding along the AC direction to reach the opposite polarization state (Fig. 1b). However, due to the abundant natural stacking faults[31], it is hard to detect the ferroelectricity in intrinsic γ-InSe. As a breakthrough, the trace Y-doping magically eliminates the stacking faults[12] and subsequently enhances the net polarization in γ-InSe (i.e., InSe:Y). Our first-principles simulations confirm that the sliding along AC direction has lower barrier than other directions. Simulations also reveal that the doped Y occupying the interstitial site (Supplementary Note 2) in the vdW-gap ($Y_i$ as an interlayer bridge; Fig. 1c) is energetically the most probable to form in γ-InSe due to the lowest formation energy, as compared to that of substituting Se ($Y_{Se}$) or In ($Y_{In}$), under both In-rich and Se-rich conditions (Fig. 1d, red lines). The role of these Y-interstitial defects is in two-fold. On one hand, $Y_i$ can restrain the

random interlayer sliding to reduce the stacking faults and stabilize the crystal structure at ambient temperature by significantly increasing the interlayer sliding barrier (Fig. 1e and Supplementary Note 3), well agreeing with the higher hardness observed after Y-doping[12]. On the other hand, the structural distortions induced by the Y-interstitial defects also lead to significant increase in the polarization (from 0.096 μC/cm² to 0.931 μC/cm²; Supplementary table 1), making the polarization more experimentally detectable. Subsequently, the robust ferroelectricity can be stabilized and observed experimentally, as representatively shown in the standard amplitude and phase hypothesis loops from PFM ("Methods", Fig. 1f, Supplementary Figs. 4–6, Supplementary Note 5), from which we find an evident negative thickness evolution of the OOP effective-$d_{33}$ piezoelectric constant ($d^{eff}_{33}$) with a saturation maximum value of ~14.0 pm/V (Supplementary Fig. 4).

### In-situ biasing-triggered interlayer sliding, ripplocation formation/motion and resulted polarization switching

We monitor the potential layer stacking changes at atomic-scale by tracing the interlayer sliding from the FIB-prepared InSe:Y cross-section specimen using the in-situ biasing Cs-TEM system (Fig. 2a and Supplementary Fig. 7a), where the specimen is adhered to the top of a copper gird (as the bottom electrode). The probe is a mobile electrode connecting to the electric measurement system, enabling the application of the external OOP electric field and the simultaneous

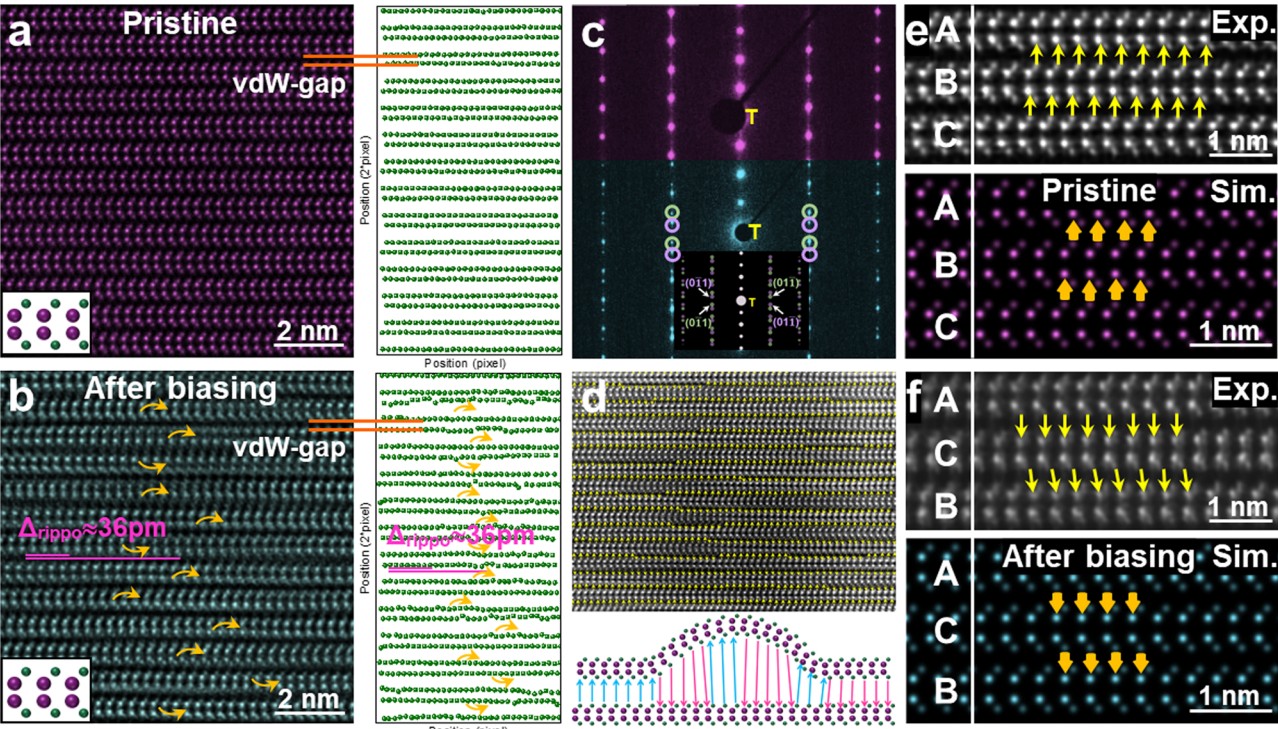

**Fig. 3 | Microstructure and polarization evolution after in-situ biasing.** HAADF-STEM images (left) and corresponding atomic positions of Se (right, identified by Calatom software) of InSe:Y before (**a**) and after (**b**) in-situ biasing process, showing obvious electric field induced ripplocation formation and motion but without any atomic rearrangement of each single layer (left, inset). The arc arrows in (**b**) roughly mark the position where the ripplocation occurs. **c** SAED patterns of InSe:Y before (upper) and after (lower) in-situ biasing process. **d** Polar map (upper) of (**d**) with the arrows only showing the polarization directions (from anion to cation) with the schematic (lower) showing the influence of the ripplocation on the polarization switching. **e, f** Atomic-level zoom-in HAADF-STEM images marked with the polarization directions (upper) in (**d**) together with the simulated ones (lower) of two unit-cells of γ-InSe with totally reversed layer stacking and polarization states.

observation of the interlayer sliding. The upper protective layer (Pt layer) in the FIB-prepared lamella is retained as the electrode for probe contact and the area for observation is in a certain distance away from the contact point to avoid the contact-induced strain or stress. During the bias sweeping from −8 V to 8 V, high-speed camera (25 frames/s) is used to record the evolution of the microstructure of InSe:Y at atomic level. The representative HRTEM snapshots (Fig. 2b) and the corresponding fast Fourier transform (FFT) patterns (Fig. 2b, inset, Supplementary Fig. 8) illustrate clear stacking changes including the obvious ripplocations and the layer-sliding after applying the voltage (e.g., at ±8 V). A rather flat layer arrangement is observed if the bias is low or off (e.g., at ±0.5 V; also see the pristine structure of InSe:Y in Fig. 3a), demonstrating the possible electric-field-dependent reversible structure changes. This phenomenon repeats as we periodically sweep the bias voltages (clearer in Supplementary Movies 1 and 2). Therefore, the ripplocation formation and motion[1,20,27] are mainly dependent upon the bias, and it may be caused by the interlayer sliding and therefore probably involve the real-time ferroelectric polarization switching, as shown in bias-dependent PFM loops (Fig .1f). Interestingly, the new diffraction patterns (marked by arrows in Fig. 2b, insets; Supplementary Fig. 8) emerge at higher biases of ±4 V and ±8 V, indicating the polarization switching of InSe:Y as simulated in Supplementary Fig. 9 and depicted in Supplementary Note 6.

Furthermore, the I-V curves during in situ TEM observation are recorded exhibiting repeatable loops with bias sweeping (Supplementary Fig. 10), on which a transition of the current occurs at the corresponding voltage possibly due to the polarization switching, also suggesting the possible connection of the external electric field induced microstructural changes in InSe:Y. Moreover, we perform C-AFM probe measurement (Methods; Fig. 2c) to examine the fine

feature of I-V curve and confirm the current and resistance character in InSe:Y, which demonstrates distinctive hysteresis features that are characteristic of obvious resistance changes during forward/reverse voltage sweeping (Fig. 2d). The measured current and associated resistance show obvious jumps with voltage increasing (Fig. 2e). The stepwise feature of resistance indicates that there are multiple polarization states[2,11,14] during polarization reversal in InSe:Y (Fig. 2f). These results provide solid evidences to confirm the link between the polarization reversal and the microstructural changes.

Limited by the influence of zone axis variations, it is difficult to get the clear HAADF atomic images during the in-situ biasing process. Therefore, in the repeated biasing circle, we randomly withdrew the probe and carried out atomic imaging analyses to examine the atomic configuration of the instantaneous state at HAADF-STEM mode. As shown in Fig. 3, we can distinguish the difference of the crystal structures peculiar to electric field induced different states. As compared to the pristine state (zero-bias) of InSe:Y showing a clear and neat arrangement of the atomic layers (Fig. 3a), atomic-scale ripplocations are observed after applying the bias (Fig. 3b). Notice that the potential twist in the layer interface can be neglected by referring to the highly-contrasted atomic imaging in each single layer. We explicitly measure the distance change of the interlayer spacing due to the ripplocations, showing an average value of ~36 pm (varying within 20 pm-50 pm), which is comparable to the reported value shown previously in transition metal dichalcogenides both theoretically and experimentally[27]. Such variation of interlayer spacing further confirms the ripplocation features (Supplementary Note 7).

Additionally, from the selected area electron diffraction (SAED) analyses, only one set of SAED pattern can be observed in the pristine γ-InSe:Y specimen (Fig. 3c, upper), but two sets of SAED patterns

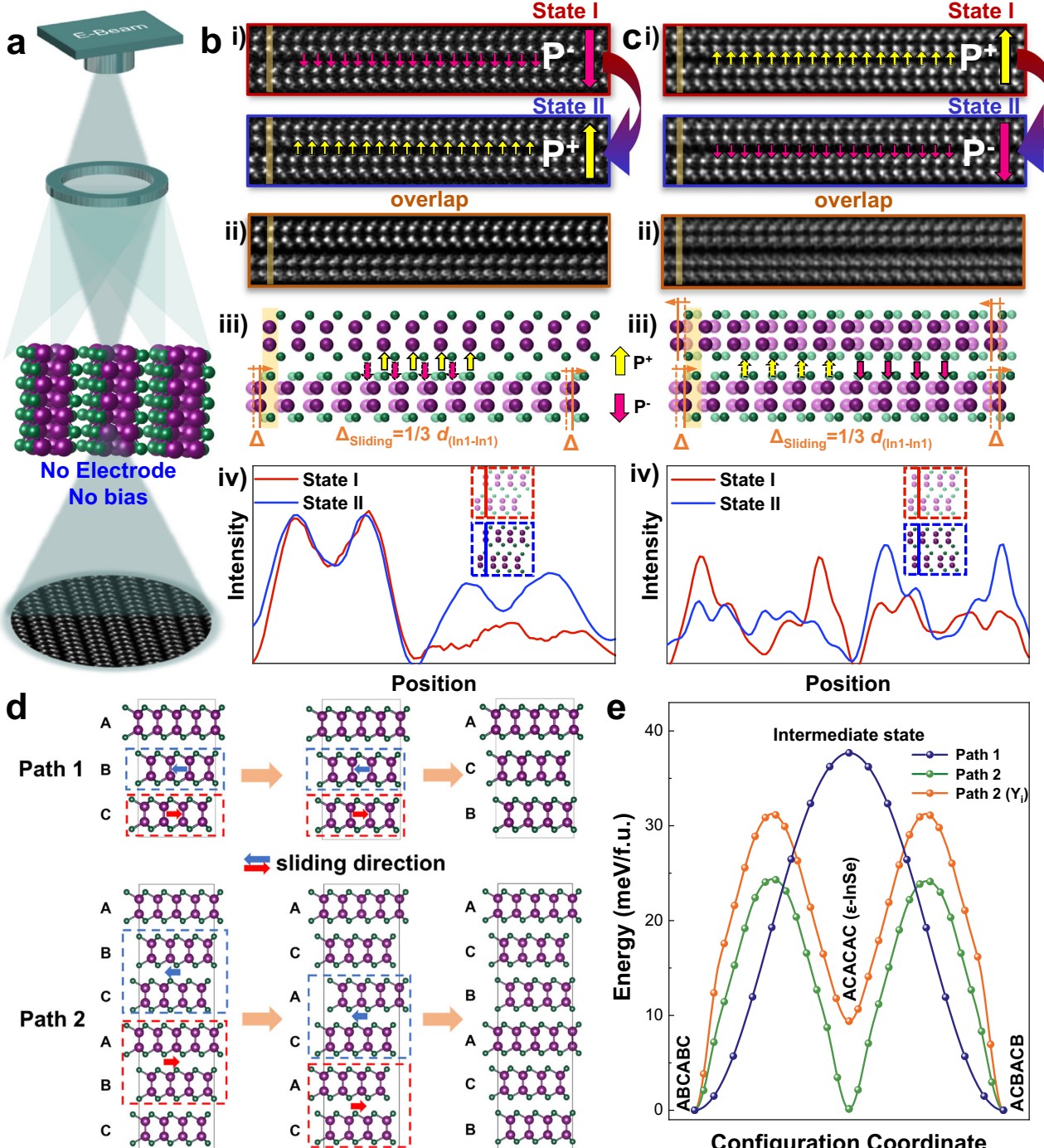

**Fig. 4 | Sliding and reversal dynamics under e-beam illumination together with the results of first-principles calculations. a** Schematic of the in-situ e-beam illumination of InSe:Y lamella. Atomic-scale single-layer (**b**) and bilayer (**c**) sliding-induced OOP polarization switching at initial (state I) and final (state II) states (i), and the overlapped states (ii) together with the structural models showing the polarization switching at initial and final states (iii) and the intensity line profiles of the initial and final states of interlayer sliding (iv). **d** Schematic for sliding paths

from one ferroelectric state (ABC) to the other bistable state (ACB); path 1 via the simultaneous forward (backward) sliding of the single B (C) layer with a 1/3 in-plane unit cell, and path 2 via the simultaneous sliding of bilayers (i.e., the metastable ACAC stacking of ε-InSe phase). **e** Calculated barriers for sliding paths 1 and 2. Notice that the results of the path 2 ($Y_i$) correspond to the case in our experiments after Y-doping.

appear after in-situ biasing, implying bias-induced reversed structural polarization (Fig. 3c, lower; Supplementary Note 6). Detailed reduced FFT analyses for the HAADF images demonstrate same FFT patterns for randomly selected regions of the pristine specimen (Supplementary Fig. 11a–c), but different FFT patterns for those of the after-biasing specimen (Supplementary Fig. 11d–i), suggesting the existence of multiple polarization states induced by the external electric field. The

multiple polarization status of InSe:Y is clearly shown in HAADF images by the polarization directions (Fig. 3d), as well as in the zoom-in HAADF images (Fig. 3e, f, upper) from Fig. 3d and the simulated ones (Fig. 3e, f, lower) that illustrate two unit-cells of γ-InSe with totally reversed layer stacking and polarization states. These results also imply that sliding-induced multiple polarization states can be modulated in multilayer 3R InSe by external biasing, as reported and mentioned above in the

ladder ferroelectricity[2] and multiple polarization states of other sliding systems[11,14], showing exciting promise in the multi-level storage.

## Atomic-level polarization switching dynamics under e-beam illumination

We should point out that (i) the interlayer sliding barrier in 2D vdW ferroelectric systems is theoretically low (Supplementary Table 2), e.g., ~0.15 meV/f.u. for bilayer $WTe_2$[17] and ~7.5 meV/f.u. for bilayer 3 R $MoS_2$ with thickness-dependence[11]; and (ii) during the Cs-(S)TEM measurements, besides the applied external electric field, the e-beam illumination also induces an electric field in nanoscale[23–25,28,29,32], whose intensity depends on the e-beam injection dose[21,29]. It has been employed to flip the ferroelectric polarization in many ferroelectric materials including the hexagonal $YMnO_3$[21] and the newly-emerging wurtzite ferroelectrics[26]. Therefore, we carry out atomic-imaging on our InSe:Y samples only under the tunable e-beam illumination conditions without the probe and any external field (Fig. 4a). We clearly observe the interlayer sliding between adjacent layers driven by the weak e-beam-induced electric field. As shown in Fig.4b, e and Supplementary Fig. 12, compared to the state I, this field makes the mainly-focused adjacent single layer or two layers slide along the AC direction (i.e., a fast interlayer sliding dynamic) by a 1/3 in-plane unit cell (Fig. 4b$_i$, c$_i$). This can be well rationalized in the overlapped images (Fig. 4b$_{ii}$, c$_{ii}$) and the intensity line profiles (Fig. 4b$_{iv}$, c$_{iv}$) of the states I and II involving the interlayer slide, which exactly leads to the opposite polarization (up and down vertically; confirmed by the atomic simulation results in Fig. 4b$_{iii}$, c$_{iii}$) (also see Fig.1b). This e-beam induced sliding behavior also provides evidence for the extra low sliding barriers between layers as predicted theoretically (Fig. 1e). The observations of one layer or two layers interlayer sliding-induced polarization switching pave the way for the potential application in ultrathin (from bulk to bi-/few-layered InSe) non-volatile memories.

Furthermore, first-principles calculations are employed to investigate the sliding-induced polarization reversal pathways along AC direction and the corresponding polarization (Supplementary Note 2). Based on the crystal structure of γ-InSe, or ABC-stacking in a unit cell, the sliding of two adjacent layers can intuitively lead to the polarization reversal (Supplementary Note 4), e.g., ABC to ACB (or path 1 in Fig. 4d, upper). The energy barrier for this path is calculated as 37.2 meV per formula unit (f.u.) (Fig. 4e). However, quite a few AC-stacking structures (or ε-InSe)[33] are experimentally observed (Supplementary Fig. 13), which cannot be explained based on path 1. Instead, a two-step path with four layers moving together at each step is proposed (path 2 in Fig. 4d, lower). In the first step, four layers form two sliding units (marked by dashed rectangles) and they slide toward each other, making ABCABC stacking shift to ACACAC stacking (or ε-InSe), whereas new sliding units form in the second step and their toward-sliding makes ACACAC stacking shift to ACBACB stacking, resulting in the polarization reversal. The energy barriers of the two steps in path 2 are calculated to be ~24.1 meV/f.u., ~34% lower than the barrier of path 1, meaning path 2 has a higher probability to happen. Importantly, this new mechanism not only perfectly explains the existence of ε-InSe in γ-InSe, but also gets directly observed by in-situ STEM (Supplementary Fig. 13). For both proposed paths in 3-layer or 6-layer unit cells, the interlayer shift is limited to the long-path sliding (Supplementary Notes 3 and 4).

For the real case with a Y interstitial (concentration corresponding to one Y atom in 192 InSe f.u.; Supplementary Fig. 14), the energy barrier of path 2 rises to ~31.0 meV/f.u., or 29% increase, and still lower than that of path 1 for the undoped InSe (Fig. 4e). These results reveal the role of the Y-doping, which not only eliminate the random interlayer sliding in pure InSe with a higher energy barrier, but also stabilize the polarization perpendicular to the InSe:Y layers for a longer range. Combined with the predicted significant increase

in OOP polarization, the robust ferroelectricity can be observed experimentally.

Along the sliding path, the OOP polarization exhibits a non-monotonic behavior, i.e., increase first, decrease to zero, and then flip to the opposite direction, whereas in-plane polarization only fluctuates without reversing its direction (Supplementary Fig. 15). We also evaluate the coupling between the in-plane polarization with the applied electric field in our experiment, and find that the coupling is at most two orders of magnitude lower than the sliding energy barrier, indicating that the influence of in-plane polarization on the OOP polarization reversal is negligible.

## Discussion

In summary, we directly visualize the in-situ interlayer sliding-induced polarization switching in vdW-layered sliding ferroelectric InSe:Y driven by either the external electric field or only the e-beam illumination of the (S)TEM system. The polarization reversal pathway prefers to the simultaneous relative sliding of two adjacent bilayers, consistent with the prediction from first-principles calculations. The sliding-induced stable and robust ferroelectric InSe:Y gives an extrapolated-saturated $d^{eff}_{33}$-value of ~14.0 pm/V, orders of magnitude higher than other 2D vdW few-layer structures, e.g., ~0.17 pm/V for monolayer α-$In_2Se_3$[34] and ~1.00 pm/V for bilayer 3R-like or 2H-like $MoS_2/WS_2$[10]. We also reveal the striking role that the slightly doped Y plays in the observation of sliding dynamics. Y-doping creates $Y_i$ in the vdW-gap, which not only optimizes the sliding barrier and enhance the even stability of layers, leading to the significant decrease of the natural stacking faults in super-flexible undoped InSe, but also raises the OOP polarization by one order of magnitude. Our results shed light on the sliding dynamics of the sliding ferroelectricity in 2D vdW-layered structures and verify the existence of multi-level polarization states in this material system, which should revolutionize the practical applications of slidetronics from ultrathin layer to semiconductors with high-speed data processing under low energy cost.

## Online content

## Methods

### Crystal growth and flake transfer

The InSe and InSe:Y crystal growth was reported in our previous work[12]. InSe and InSe:Y flakes for SHG and PFM analyses were prepared by employing a scotch-tape mechanical exfoliation method. To eliminate the interface effect, multilayer graphene was introduced on the highly-doped silicon substrate before InSe:Y flakes were transferred.

### PFM and electrical measurements

Dual amplitude resonance tracking (DART) mode PFM measurements were carried out on an AFM system (Asylum Research Cypher, Oxford, UK). The spring constant of 2.0 N/m for the conductive Pt-coated silicon tip was adopted with the detected AC voltage of 0.5 V and the tip-sample contact resonant frequency of ~220 kHz. In addition, domain writing was performed by using the LithoPFM mode of the Asylum Research software with reverse DC biases ($V_{dc} = \pm 5$ V).

C-AFM I−V curves were obtained with bias sweeping from the positive to negative voltage circle, like 0 V to +1.5 V to −1.5 V to 0 V, with a time period of 1 s.

### In-situ biasing Cs-TEM

TEM samples were prepared on a dual-beam FIB system (Helios G4 UX, FEI, USA).

The atomic-scale in-situ biasing TEM analyses were performed on a Cs-TEM (JEOL Grand ARM300, Japan) with a PicoFemto double tile biasing TEM holder (ZEOTools Technology Company) at HRTEM mode. A tungsten tip was used as the mobile electrode, which was precisely controlled by a piezoelectric system. The in-situ videos were obtained using Gatan OneView camera with 4 k resolution by a speed of 25 frames/s. The in-situ video was drift-corrected and processed by using GM3 in-situ data processing. The HRTEM snapshots corresponding to different external voltages were extracted from the in-situ video and filtered to remove the high-frequency noise (Supplementary Movies 1 and 2).

Atomic-resolution imaging measurements before and after biasing were performed on the Cs-TEM at the AC-HAADF-STEM mode. Atomic-scale polarization switching under e-beam illumination was observed with the spot size of 8 C with a probe current of 23 pA or 6 C with 30 pA.

The simulated HAADF-STEM images were performed using Mac-Tampas software. The simulated SAED patterns were obtained using CrystalKitX software.

Detailed atomic arrangement analyses for HAADF-STEM images combined with atomic quantification were performed using the Cala-tom software based on custom MATLAB scripts.

### DFT calculations

The first-principles were carried out using the density functional theory (DFT) as implemented in the Vienna ab initio simulation package (VASP)[35,36]. The Perdew–Burkes–Ernzerhof (PBE)[37] form of the generalized gradient approximation (GGA) to the exchange-correlation functional was adopted. The projector augmented-wave (PAW) pseudopotentials with 520 eV cutoff for the planewave basis set were used[38]. Optb86b-vdW form is selected to describe the vdW interaction[39]. A 192-atom supercell is built to calculate the Y-doped defect properties. The energy barrier for sliding was realized using the nudged elastic band (NEB) method[40,41].

For defect properties, the formation energy of a defect $\alpha$ in the charge state $q$ is calculated as[42],

$$\Delta H_f(\alpha, q, E_F) = E(\alpha, q) - E_{host} + \Sigma n_i(E_i + \mu_i) + q(E_F + E_{VBM} + \Delta V) \quad (1)$$

where $E_{host}$ is the total energy of the defect-free supercell, and $E(\alpha, q)$ is the energy of the supercell with a defect $\alpha$ with charge state $q$. The chemical potential of element $i$, which is denoted as $\mu_i$, is calculated with reference to the energy of its elemental phase $E_i$ per atom. $\mu_i = 0$ means that the element $i$ is so rich that its pure elemental phase can form. $E_F$ is the Fermi level referenced to the valance band maximum $E_{VBM}$ of the defect-free structures. $\triangle V$ includes the electrostatic potential alignment of $E_{VBM}$ as well as the finite-size correction.

### Reporting summary

Further information on research design is available in the Nature Portfolio Reporting Summary linked to this article.

## Data availability

All data are available in the main text or the Supplementary Material.

## Code availability

The codes used to process and analyze the data presented in this work are available from the corresponding author upon request.

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

## Acknowledgements

This work was financially supported by the National Key Research Project of China grant 2022YFA1402902 (CGD), the National Natural Science Foundation of China (NSFC) grants 12134003 (CGD), 62274061 (FYY) and 61974042 (RH), the Natural Science Foundation of Chongqing, China grants CCSTB2023NSCQ-MSX0975 (RJQ) and the Fundamental Research Funds for the Central Universities grant YBNLTS2013-019 (FRS). M.J. acknowledges the 'Shuguang Program' supported by Shanghai Education Development Foundation and Shanghai Municipal Education Commission. M.J. and X.L. acknowledges project supported by the Space Application System of China Manned Space Program.

## Author contributions

R.Q. and F.Y. formulated and supervised the project. M.J., X.L. and F.S. prepared the samples. F.S., R.Q., B.L., R.G., Y.Z. and R.H. conducted the TEM investigations and characterizations. Y.W. Z.L. and H.L. performed the DFT calculations. F.S., R.Q., F.Y., M.W. and C.D. analyzed the data. F.Y. and R.Q. drafted the original manuscript. J.M., X.L., M.W., J.C. and C.D. reviewed and edited the manuscript with contributions from all authors.

## Competing interests

The authors declare no competing interests.
