## [Peer Review File · Nature Communications]

Atomic-level polarization reversal in sliding ferroelectric semiconductorsEditorial Note: This manuscript has been previously reviewed at another journal that is not operating a transparent peer review scheme. This document only contains reviewer comments and rebuttal letters for versions considered at *Nature Communications*.

REVIEWER COMMENTS

Reviewer #1 (Remarks to the Author):

I have previously reviewed this manuscript, and after a number of changes it has improved, however certain issues still remain.

1. Beam-induced switching:

"Furthermore, local electrostatic charging during the electron-beam (e-beam) illumination (without the electrode/probe), which can induce the necessarily large local electric field^{20,24} for polarization switching in ferroelectrics^{25,27,28}, enables clear observation of the kinetics of atomic-scale interlayer sliding in InSe:Y, confirming the OOP polarization switching dynamics, as technically the effects from the electrode (or charging) and the local mechanically-induced strain are excluded."

This sentence and further text, being unnecessary complex, also presumes that in-situ switching observed in the lamella would be similar to that in planar devices. I would like to point out that lamella sample preparation is extremely invasive, and to my experience just the act of cutting out a slice of sample often produces large strain in the specimen, an issue well known in the field. This strain frequently relaxes via formation of mechanical defects during the imaging. Sometimes, the strain is so significant, that the heterostructures split open during thinning down even before the (S)TEM imaging. It is simply misleading to presume such samples have no strain, and to say this behaviour has anything to do with switching "kinetics" as a general property of the material. My overall feeling reading the manuscript is that authors either have little understanding of the system they are studying, or so focused on overselling it, they make otherwise solid experimental work look worse.

2. Looking at the probe-induced switching:

I appreciate the addition of the tunnelling current dependency vs bias applied. This is however only single one-way trace and interpretation of small bumps on a single curve in this scenario is not reliable to put it mildly. Interpretation of tunnelling characteristics on a sample that virtually demolished by charged ions seems very dubious. This is not the direction I was expecting authors would take – instead it would be much better to demonstrate the switching behaviour while providing a solid proof that the probe is not in mechanical contact with the sample, something that can be easily established by taking a larger field of view image with the probe (during the biasing) and the sample, repeating this control during each change of bias voltage.

As it stands, the authors bring the probe in to apply bias and then withdraw it when imaging. The probe is clearly in mechanical contact during the bias sweep, because they are able to record the I(V) curve. It can, therefore, mechanically influence the sample and the issue is the same as the last round – switching can be either due to the mechanical force applied by the probe or the E field. Then the authors withdraw the tip and visualise the changes and to my great surprise claim they excluded effects of strain.

"Limited by the electron microscopy resolution and the influence of zone axis variations, it is difficult to get the clear HAADF atomic images during the in-situ biasing process. Therefore,

in the repeated biasing circle, we randomly withdrew the probe and carried out atomic imaging analyses to examine the atomic configuration of the instantaneous state at HAADF-STEM mode. This can rule out the effects of charging and potential strain on the local polarization switching.”

In summary, I would recommend to

(1) rewrite the text to make it more rigorous and careful with claims made. Improvement and simplification of the English would also help.

(2) clearly state that lamellae studied are not an ideal representation of processes in bulk samples and explain the invasiveness of their fabrication, and therefore residual strain and its consequences.

(3) reproduce the biasing experiment with the probe not touching the sample, with evidence of this presented.

Reviewer #2 (Remarks to the Author):

See attached.

After the first round of reviewing, the authors have diligently addressed my concerns, incorporating additional experiments that significantly enhance the emphasis on the interlayer shift-induced polarization switching under the influence of an electric field. The study is timely and poised to garner substantial interest in the condensed physics community. I recommend considering two minor issues before final acceptance:

1. While the authors assert that the polarization switching mechanism is “ripplocation formation and motion”, a point of contention among all referees, they provide compelling evidence to support their claim. From my perspective, the electric field may induce instability at the interface, potentially resulting in the enlargement of interlayer spacing or interlayer twist. The second referee has raised concerns about interlayer twist, and it is essential to address this aspect. The authors are suggested to provide insights into the absence of twist within the interface, or include discussions on alternative polarization switching mechanisms involving interlayer twist.

2. Regarding the calculations of the sliding energy barrier, the authors considered two routes: relative sliding and single layer sliding. The current representation may pose challenges for readers to comprehend at first glance. To enhance clarity, I suggest the authors re-plot Fig. 1e, incorporating a complete path that includes both routes (i.e., shift from both left and right from AB stacking mode) and represents the real coordinates (e.g., in terms of $d_{(In1-In1)}$). This will enable a clear definition of relative sliding and single layer sliding on the revised Fig. 1e.

The above discussion suggests that relative sliding should not be considered the preferred reversal pathway (due to the cross of the high energy state AA stacking mode), as the preferable path is single layer sliding. This contradicts the assertions made on pages 5, 10, and 11. For instance, in path 1 of Fig. 4h, the path involving the shift of the middle layer by $1/3d_{(In1-In1)}$ and the bottom layer by $2/3d_{(In1-In1)}$ along the left side should have a smaller energy barrier than that of path 1.

Comparing the energy barrier of path 2 and path 1 becomes intricate due to the difference in layer count—one with three layers and the other with six layers. In the case of six layers, numerous routes (and meta-states) are available to transition from full polarization up to full polarization down, one of which is illustrated in path 2. I presume the authors limit the interlayer shift pathway to $1/3d_{(In1-In1)}$, although this is not explicitly noted in the main text. Additionally, the authors use six layers as an example to explore the energy barrier, representing a typical case reported in experiments (which is multilayer). To enhance clarity, it would be beneficial to add a clarification, such as "for the six-layer system, the interlayer shift pathway is limited to $1/3d_{(In1-In1)}$," in relevant discussions, for instance, on page 5, line 15.

Response letter

We thank the referees very much again for their enlightening and helpful comments on our revised manuscript (#NCOMMS-24-01933-T). On the light of the referees' comments, we are glad to demonstrate our recent achievements to reply their concerns, as listed below point-by-point.

Reviewer #1:

I have previously reviewed this manuscript, and after a number of changes it has improved, however certain issues still remain.

Q1. Beam-induced switching:

"Furthermore, local electrostatic charging during the electron-beam (e-beam) illumination (without the electrode/probe), which can induce the necessarily large local electric field^{20,24} for polarization switching in ferroelectrics^{25,27,28}, enables clear observation of the kinetics of atomic-scale interlayer sliding in InSe:Y, confirming the OOP polarization switching dynamics, as technically the effects from the electrode (or charging) and the local mechanically-induced strain are excluded."

This sentence and further text, being unnecessary complex, also presumes that in-situ switching observed in the lamella would be similar to that in planar devices. I would like to point out that lamella sample preparation is extremely invasive, and to my experience just the act of cutting out a slice of sample often produces large strain in the specimen, an issue well known in the field. This strain frequently relaxes via formation of mechanical defects during the imaging. Sometimes, the strain is so significant, that the heterostructures split open during thinning down even before the (S)TEM imaging. It is simply misleading to presume such samples have no strain, and to say this behavior has anything to do with switching "kinetics" as a general property of the material. My overall feeling reading the manuscript is that authors either have little understanding of the system they are studying, or so focused on overselling it, they make otherwise solid experimental work look worse.

Response: We sincerely appreciate you for the helpful comments. It is indeed a difficult yet critical concern.

FIB is one of the critical techniques for material microstructural researches, especially for the TEM sample preparation or patterning. It is well recognized that ion implantation and material amorphization during FIB processing would be introduced into the target material and could generate stress or strain, which may cause the structural deformation of the target material. In fact, the “undesirable” defects introduced during FIB fabrication process can be largely reduced, which will not influence the subsequent TEM observations by referring to the technique keys, e.g.,

- 1) Attaching the liftout TEM sample to the head of grid so that the strain can be maximally released (Fig. R1a-c);
- 2) A more even lamella thickness can be achieved by slightly tilting the sample during the milling process to produce a wedge geometry, followed by a fine milling step at the original milling angle (Fig. R1d-f);
- 3) Using low kV cleaning or subsequent ion milling by applying precision ion polishing system (PIPS) to reduce the material amorphization;
- 4) Using the lower processing ion current to reduce possible strain; and
- 5) Reducing the processing area.

Fig. R1. Optimizing the FIB fabricating procedures for high-quality TEM samples.

The FIB instrument used in this study is a dual beam Helios G4 from FEI company, an innovator and leading supplier of solutions for the global nanotechnology community,

with low voltages modules (30 kV - 2 kV). In this setup, the FIB column and SEM are integrated on a platform, and the two optical axes converge at the same point on the sample surface, enabling efficient in-situ observation, non-destructive control, and real-time monitoring of the FIB manufacturing process. Based on our experience, with carefully operation, the strain in the specimens can be largely relaxed and reduced, which will not influence (S)TEM observation and diagnosis. From the bright field TEM image in Fig.R1f, you can see that the FIB-prepared TEM sample can be thin with uniform thickness and no obvious strain (with same diffraction contract).

For the beam-induced switching, we have carried out a series of comparable examinations on many FIB-fabricated undoped InSe lamellae and ferroelectric InSe:Y lamellae. It is interesting to point out that the beam-induced atomic switching can only be detected in InSe:Y samples, while it cannot be observed in undoped InSe that has no detectable ferroelectricity. Such observation implies that the switching should not be attributed to the residual stress relaxing during TEM observation.

Following your suggestions, we have revised this sentence and following text as well as other related parts by removing the arbitrary claiming of no strain during the preparation process of the FIB samples, in order not to make misunderstanding.

Q2. Looking at the probe-induced switching:

I appreciate the addition of the tunnelling current dependency vs bias applied. This is however only single one-way trace and interpretation of small bumps on a single curve in this scenario is not reliable to put it mildly. Interpretation of tunnelling characteristics on a sample that virtually demolished by charged ions seems very dubious. This is not the direction I was expecting authors would take – instead it would be much better to demonstrate the switching behavior while providing a solid proof that the probe is not in mechanical contact with the sample, something that can be easily established by taking a larger field of view image with the probe (during the biasing) and the sample, repeating this control during each change of bias voltage. As it stands, the authors bring the probe in to apply bias and then withdraw it when

imaging. The probe is clearly in mechanical contact during the bias sweep, because they are able to record the $I(V)$ curve. It can, therefore, mechanically influence the sample and the issue is the same as the last round – switching can be either due to the mechanical force applied by the probe or the E field. Then the authors withdraw the tip and visualize the changes and to my great surprise claim they excluded effects of strain.

“Limited by the electron microscopy resolution and the influence of zone axis variations, it is difficult to get the clear HAADF atomic images during the in-situ biasing process. Therefore, in the repeated biasing circle, we randomly withdrew the probe and carried out atomic imaging analyses to examine the atomic configuration of the instantaneous state at HAADF-STEM mode. This can rule out the effects of charging and potential strain on the local polarization switching.”

Response: We appreciate your precious suggestions.

First, upon this concern “Interpretation of tunnelling characteristics on a sample that virtually demolished by charged ions seems very dubious”, we would like to mention that InSe:Y sample in this work is a ferroelectric material as we have previously reported in *Nature Communications*, 2023, 14, 36. The flakes used for CAFM measurements were first checked by using PFM mode to confirm the ferroelectricity. Then, the CAFM mode was adopted to obtain the tunneling characteristic on the same sample using the same instrument. Furthermore, to eliminate the influence of the charging ions at the interface of the substrate and our 2D InSe:Y, thin graphene was introduced under our InSe:Y flake.

For your concerning on the strain effect during in-situ biasing TEM observation:

During the in-situ biasing TEM investigation, the proceeding of the probe is controlled by piezoelectric module with a step of 0.4 nm in X, Y directions and 0.04 nm in Z direction, to make the contact of the probe with the sample more tunable. In Video R1, we have shown the verification experiments on control of the contact force between probe and sample. We demonstrate the process of the forward and backward probe to the top surface of the FIB-fabricated lamellae, clearly exhibiting how we

accurately tune the contact of the probe with the sample to reduce the possible mechanical strain or stress. Beneficial from the high stability of the in-situ biasing system (high-quality HRTEM and atomic image can be obtained with this in-situ biasing systems), the mechanical force can be accurately controlled during the in-situ observations, which will not lead to further switching in the sample. On the other hand, the changes due to the small mechanical strain can't be retained, which will return to the pristine status after strain unloading, while the ferroelectric polarization switching is retainable, which can be observed even after the withdrawing of the probe, as presented in the manuscript (Fig.3).

Moreover, actually, in our previous experiments and new experiments, the observed areas were indeed a certain distance away from the position of the probe contact (Fig.R2), trying the best to avoid the potential probe-induced mechanical strain.

By following your suggestions, again we prepared new in-situ TEM samples, specifically by reserving the upper metal layer (Pt/W) as the electrode for probe contact (Fig.R2a, inset EDS maps showing the distribution of Pt/W at the top of InSe:Y sample), making sure that the probe does not directly contact the observed InSe:Y sample. Successfully, as expected, we get the repeatable cycled I-V loops (with obvious hysteresis characteristics) (Fig.R3) for different samples or at different contact positions, similar to that in CAFM measurements and similar microstructural changes (Fig.R2c, Fig.R4) as demonstrated in our manuscript (Fig.2), which is definitely related to the ferroelectric switching due to the electric field. Same characterization techniques have been extensively used, e.g., see *Nature*, 2023, 613, 656-661 and *Sci. Adv.*, 2022, 8, eabo0773. These results provide solid evidences to confirm the link between the polarization reversal and the microstructural changes.

Fig. R2. **a**, TEM image of the optimized FIB-prepared sample with Pt/W as the top electrode. **b**, Voltage sweeping. **c - e**, HRTEM snapshots during in-situ biasing TEM analyses with voltage sweep, showing similar microstructural changes at different areas due to electric field induced polarization switching in ferroelectric InSe:Y.

Fig. R3. Corresponding I-V loop obtained during in-situ TEM observations in Fig.R2.

Video R1 (double click to activate for playing). In-situ HRTEM video obtained by using high speed CCD during voltage sweeping.

Fig. R4. Atomic imaging analyses after in-situ biasing.

In summary, I would recommend to

(1) rewrite the text to make it more rigorous and careful with claims made. Improvement and simplification of the English would also help.

Response: We are grateful for your comments on our scientific and language expressions. We have made significant revisions to the text. All main revisions have been marked in red in the revised manuscript.

(2) clearly state that lamellae studied are not an ideal representation of processes in bulk samples and explain the invasiveness of their fabrication, and therefore residual strain and its consequences.

Response: We have modified the descriptions about the lamellae fabrication in the revised version (including the main text and supplementary information). The main points are i) deleting the arbitrary assumption of the potential strain in the FIB-fabricated lamellae, although our further experimental achievements can exclude its influence on the polarization switching; and ii) adding the new I-V loops of new-fabricated samples on the in-situ biasing TEM system by following your suggestions, indeed which can intrinsically improve the effectivity and practice of the lamellae for polarization switching observation with uniform external electric field but without stain-induced influence factors.

(3) reproduce the biasing experiment with the probe not touching the sample, with evidence of this presented.

Response: Following your enlightening suggestions, new in-situ TEM samples specifically by retaining the upper protective layer as the electrode for probe contact are fabricated (Fig.R2a). During the measurements, the probe is force-optimized and kept stable to contact the electrode without direct contact with the sample. The atomic-level observation for the switching is performed at different locations with a certain distance away from the probe (Fig.R2a). We obtain not only the repeatable I-V loops (Fig.R3) but also repeatable microstructural changes (Fig.R4) in the in-situ biasing TEM system with bias sweeping, well consistent with that of CAFM

measurements and previous TEM results presented in our manuscript.

To summarize, based on new experimental results, considering the structural and polarization retention behavior of the ferroelectric materials (CAFMs loop, in-situ I-V loop, atomic imaging after electric field withdrawal) and the stability of the in-situ system, we confirm that the atomic level polarization switching is due to the E field. We hope these comprehensive results can fully address your concerns.

Reviewer #2:

After the first round of reviewing, the authors have diligently addressed my concerns, incorporating additional experiments that significantly enhance the emphasis on the interlayer shift-induced polarization switching under the influence of an electric field. The study is timely and poised to garner substantial interest in the condensed physics community. I recommend considering two minor issues before final acceptance:

Q1. While the authors assert that the polarization switching mechanism is “ripplocation formation and motion”, a point of contention among all referees, they provide compelling evidence to support their claim. From my perspective, the electric field may induce instability at the interface, potentially resulting in the enlargement of interlayer spacing or interlayer twist. The second referee has raised concerns about interlayer twist, and it is essential to address this aspect. The authors are suggested to provide insights into the absence of twist within the interface, or include discussions on alternative polarization switching mechanisms involving interlayer twist.

Response: Thanks a lot for your suggestions. Indeed, as you mentioned, there are some changes of interlayer spacing due to the electric field, as shown in Fig.3b (left and right). However, we want to emphasize that the atomic arrangements from the side-view perspective in the STEM-TEM images are clear and ordered in a large observation scale, which should suggest that there are no interlayer twists induced by the electric field. Related discussions or descriptions have been added into the revised manuscript, e.g., see page 8, line 23 - 25.

Q2. Regarding the calculations of the sliding energy barrier, the authors considered two routes: relative sliding and single layer sliding. The current representation may pose challenges for readers to comprehend at first glance. To enhance clarity, I suggest the authors re-plot Fig. 1e, incorporating a complete path that includes both routes (i.e., shift from both left and right from AB stacking mode) and represents the real coordinates (e.g., in terms of $d(I_{n1}-I_{n1})$). This will enable a clear definition of relative sliding and single layer sliding on the revised Fig. 1e.

Response: We thank you for raising this very interesting point. The sliding paths considered in Fig.1e only contain the shortest route to realize the polarization reversal (short path). It is also possible that the sliding happens towards the opposite direction and reverse the polarization, although the path is longer (notated as long path).

Following your suggestion, we have calculated the barriers for sliding in both directions and re-plotted Fig.1e (also Fig.R5). The shaded area is included in the old Fig.1e. It is interesting that, before Y doping, the one-layer sliding prefer the short path because of lower barrier (brown curve), while the relative sliding seems to have similar barrier along both directions (blue curve). This is because the relative sliding along both directions need to overcome a barrier corresponding to the “AA stacking”, and one-layer sliding only needs to overcome this barrier along the long path.

Fig. R5. The influence of doped-Y on the interlayer sliding energy barriers for bilayer InSe. Both sliding directions are included. The left part is the long path, and the right shaded part is the short path.

With Y interstitial, both one-layer sliding and the relative sliding shows a preference on the sliding direction, i.e., the short path has lower barriers than the long path for both one-layer sliding (purple curve) and relative sliding (red curve). To mimic the sliding in the bulk system, the interlayer vdW gap is maintained during the sliding. For the one-layer sliding (purple curve), the higher barrier along the long path comes from the AA stacking with the Y atom sitting in the vdW gap, while there is no AA stacking position along the short path. For the relative sliding (red curve), although the barriers along both paths have the AA-stacking, the position of the Y atom is different for the short and long paths at the barrier tops. At the barrier top of the short path, the Y atom is at the hollow site (Fig.R6a), while at the barrier top of the long path, the Y atom is sitting in between two In atoms from upper and lower layers (Fig.R6b). Such difference causes the barrier is higher along the long path than the short path for relative sliding. Based on this finding, we assume that the sliding prefers the short path with Y doping.

Fig. R6. Side-view of the Y-doped bilayer system correspond to the barrier top of relative sliding along (a) the short path and (b) the long path (two maxima of the red curve in Fig.R5).

In the revised manuscript, we have updated the Fig.1e and the relative discussions in Supplementary Notes 3 and 4.

Q3. The above discussion suggests that relative sliding should not be considered the preferred reversal pathway (due to the cross of the high energy state AA stacking

mode), as the preferable path is single layer sliding. This contradicts the assertions made on pages 5, 10, and 11. For instance, in path 1 of Fig. 4h, the path involving the shift of the middle layer by $1/3d_{(In1-In1)}$ and the bottom layer by $2/3d_{(In1-In1)}$ along the left side should have a smaller energy barrier than that of path 1.

Response: We appreciate you for this helpful comment. The purpose of Fig.1e is to demonstrate that the introduction of Y atoms in the vdW gap of a bilayer system would increase the barrier for the layer sliding and hence help to stabilize the structure. We meant to compare the barriers for relative sliding with and without Y-doping, and compare those for one-layer sliding with and without Y-doping. You carefully examined our figure and raised that the one-layer sliding generally has lower barrier than relative sliding, so the relative sliding modes that we used to demonstrate the sliding mechanisms in bulk InSe may not be the energetically favorable.

Firstly, we need to emphasize that such contradiction does not exist. This confusion comes from the difference of the sliding in the two-layer system and the bulk. In a bilayer system, the one-layer sliding does have lower barrier than the relative sliding, simply because the relative sliding has to overcome a barrier that corresponds to the AA-stacking, whereas the one-layer sliding can avoid the AA-stacking in the $1/3 d_{(In1-In1)}$ direction (as shown in Fig.R6). The AA-stacking makes the higher barrier for the relative sliding than one-layer sliding, as we explained in the reply for the previous comment. However, in the bulk system, the one-layer sliding would always hit “AA-stacking” situations, because the moving layer is sandwiched by two neighboring layers and AA stacking always forms if only one layer slides. For example, in a unit cell with three ABC-stacking layers, if we move one single layer (say layer B), sliding of B would inevitably hit A or C depending on the direction, leading to the “AA-stacking” with neighboring layers. Therefore, the one-layer sliding does not hold advantage over the relative sliding in terms of the energy barrier.

You also proposed a possible sliding path for the polarization reversal that the middle layer by $1/3 d_{(In1-In1)}$ and the bottom layer by $2/3 d_{(In1-In1)}$. We carried out a NEB calculation, and obtained a barrier of 44.5 eV per f.u., which is higher than the 37.2 eV per f.u. of the relative sliding as we adopted. At the barrier top, the third layer forms

AA stacking with the layer A in the unit cell below. At the same time, the middle layer is in between B and C. Overall, the barrier is higher than the relative sliding of the relative sliding as we proposed.

In the revised version, we made clearer descriptions about the sliding dynamics. See page 11, line 14 – 16; and Supplementary Note 4.

Q4. Comparing the energy barrier of path 2 and path 1 becomes intricate due to the difference in layer count—one with three layers and the other with six layers. In the case of six layers, numerous routes (and meta-states) are available to transition from full polarization up to full polarization down, one of which is illustrated in path 2. I presume the authors limit the interlayer shift pathway to $1/3 d_{(In1-In1)}$, although this is not explicitly noted in the main text. Additionally, the authors use six layers as an example to explore the energy barrier, representing a typical case reported in experiments (which is multilayer). To enhance clarity, it would be beneficial to add a clarification, such as "for the six-layer system, the interlayer shift pathway is limited to $1/3 d_{(In1-In1)}$," in relevant discussions, for instance, on page 5, line 15.

Response: We are grateful to you for pointing out this insufficiency. Firstly, we need to emphasize that the comparison of the energy barrier is valid because the barriers are all presented in the units of meV per formula unit. Unlike the conventional ferroelectrics, the layer sliding has much lower energy barrier. As a result, there could exist other possible reversal sliding paths. As you assumed, the sliding mechanism that we proposed is based on the experimental observation of ϵ -phase InSe, and we limit the interlayer shift pathway to $1/3 d_{(In1-In1)}$, or the short path as defined in the reply to previous two comments. To clarify this point, we explicitly add the contents as you suggested, "For both proposed paths in 3-layer or 6-layer unit cells, the interlayer shift is limited to the shortest sliding path (Supplementary Notes 3 and 4)." on page 11, line 14.

REVIEWERS' COMMENTS

Reviewer #1 (Remarks to the Author):

I'm happy with the changes made at the last round and recommend this manuscript for publication.

Reviewer #2 (Remarks to the Author):

My comments have been addressed and the manuscript is now suitable for publication.

Response Letter

Reviewer #1:

I'm happy with the changes made at the last round and recommend this manuscript for publication.

Response: We thank you very much for this positive response to our previous revisions.

Reviewer #2:

My comments have been addressed and the manuscript is now suitable for publication.

Response: We thank you very much for this positive response to our previous revisions.